

# Pain neuroscience education on YouTube

Lauren C. Heathcote[1], Joshua W. Pate[2], Anna L. Park[1], Hayley B. Leake[3], G. Lorimer Moseley[3], Corey A. Kronman[1], Molly Fischer[1], Inge Timmers[1] and Laura E. Simons[1]

[1] Anesthesiology, Perioperative, and Pain Medicine, Stanford University School of Medicine, Stanford, CA, United States of America
[2] Faculty of Medicine and Health Sciences, Macquarie University, Sydney, New South Wales, Australia
[3] School of Health Sciences, University of South Australia, Adelaide, South Australia, Australia

## ABSTRACT

**Objectives**. The Internet in general, and YouTube in particular, is now one of the most popular sources of health-related information. Pain neuroscience education has become a primary tool for managing persistent pain, based in part on the discovery that information about pain can change pain. Our objective was to examine the availability, characteristics, and content of YouTube videos that address the neuroscience of pain.
**Methods**. We conducted a systematic review of videos on YouTube using the search terms "pain education", "what is pain", and "pain brain" in January 2018. Videos were included if they were in English, were under 10 minutes long, and included information on the neuroscience of pain. Videos were coded for (i) descriptive characteristics (e.g., number of views, duration on YouTube), (ii) source and style, (iii) whether or not they addressed seven pre-determined target concepts of pain neuroscience education (e.g., 'Pain is not an accurate marker of tissue state'), and (iv) how engaging they were.
**Results**. We found 106 unique videos that met the inclusion criteria. The videos ranged from having four views to over five million views ($Mdn = 1,163$ views), with the three most highly viewed videos accounting for 75% of the total views. Animated videos were much more highly viewed than non-animated videos. Only a small number of videos had been posted by a clearly-identifiable reputable source such as an academic or medical institution (10%), although a number of videos were posted by healthcare professionals and professional medical societies. For a small number of videos (7%), the source was unclear. We found 17 videos that addressed at least one target concept of pain neuroscience science education, only nine of which were considered to be at least somewhat engaging. The target concept 'Pain is a brain output' was considered to be well addressed by the most videos ($N = 11$), followed by 'Pain is a protector' ($N = 10$). We found only one video that adequately addressed all seven target concepts of pain neuroscience education.
**Discussion**. YouTube contains a variety of videos that practitioners, patients, and families may view to access pain neuroscience education information. A small portion of these videos addressed one or more target concepts of pain neuroscience education in an engaging manner. It is yet to be determined to what extent patients are able to learn information from these videos, to what extent the videos promote behavior change, and thus to what extent the videos may be useful for clinical practice.

Corresponding author
Lauren C. Heathcote,
lcheath@stanford.edu

# INTRODUCTION

Since its inception three decades ago, the Internet has changed the way that consumers access health care information. Individuals with chronic illnesses, in particular, are increasingly using online resources to manage their conditions (*Fox, 2006*) and to help them make health care decisions (*McMullan, 2006*). Some patients even report feeling more confident in the information they access online than the information provided by their physician (*Diaz et al., 2002*).

YouTube is the second most popular website in the world (*Alexa, 2018*; Top 500 websites on the internet") and is a primary Internet platform for consumer-targeted health information (*Sampson et al., 2013*). The video-sharing website provides a medium for delivery of engaging, multimodal information that can be freely uploaded by industry, government and non-government organizations, as well as health care providers and consumers themselves. Assessing the availability and accuracy of online health information, including on YouTube, is a rapidly developing area of study for health care researchers (*Sampson et al., 2013*).

The availability and content of online health education is particularly pertinent when it comes to pain. Persistent pain reduces quality of life in up to one in five adults, and is one of the most burdensome health conditions in terms of years lived with disability (*Vos et al., 2016*; *James et al., 2018*) and economic cost (*Gaskin & Richard, 2012*). Persistent pain is also common and disabling in youth (*Huguet & Miró, 2008*; *King et al., 2011*), a demographic that is growing up in the era of technology and has far surpassed older age groups in terms of Internet use (*Wartella et al., 2016*). In 2018, the International Association for the Study of Pain (IASP) held its 'Year for Excellence in Pain Education'. This recognizes that: (1) education is a recommended first-line treatment for persistent pain in guidelines internationally (*Buchbinder et al., 2018*; *Foster et al., 2018*; *Hartvigsen et al., 2018*) although there are caveats (*Moseley, 2018*), (2) substantial progress in pain neuroscience over the last 40 years has rendered the common understanding of persistent pain inaccurate (*Moseley & Butler, 2015*), and (3) contemporary pain neuroscience education has become a central component of pain treatment worldwide. Seminal work by Fordyce (*Fordyce, 1984*) led to early psychoeducation programs for persistent pain focusing on how to manage and cope with pain. More recently pain education has focused instead on scientific concepts that underlie the experience of pain; they aim to explain what pain is, what function it serves, and what biological processes are thought to underpin it (*Moseley & Butler, 2015*). These have been neatly summarized as the 'what, why, and how of pain' (J Pate, T Noblet, J Hush, M Hancock, M Pounder, R Sandells, V Pacey, 2018, unpublished data). There is now Level 1 evidence that this kind of pain education has a range of clinically important effects, including reduced pain and disability, reduced catastrophizing, increased self-efficacy, and enhanced participation in biopsychosocially-based rehabilitation, with, critically, no identified harms

or side-effects (*Moseley & Butler, 2015*; *Louw et al., 2016*). The rapid progress in this field is reflected in the multiple terms now used to describe pain education, including 'pain neuroscience education', 'pain biology education', 'therapeutic neuroscience education', 'neurophysiological pain education', and 'Explaining Pain' (*Robins et al., 2016*). In this paper, we will refer to these approaches broadly as pain neuroscience education.

Despite the importance of high-quality pain neuroscience education and the reality that the majority of people with access to the Internet use it as a primary source of health education, there seems to have been no prior systematic attempt to evaluate the quality of YouTube-based pain neuroscience education. A small number of studies (e.g., *Corcoran et al., 2009*; *Bailey et al., 2013*) have reviewed the websites arising from commonly used pain-related search terms (e.g., *chronic pain*) entered into popular search engines. These studies have typically found that websites range substantially in quality, with some providing accurate and detailed information about pain but most providing incomplete and incorrect information. Other studies have reviewed the availability of smartphone applications for pain management (*Lalloo et al., 2015*), finding that the most common content is pain self-care and, importantly, that most applications lack involvement of health care professionals in their development. We aimed to build on these studies by examining the availability, characteristics, and content of pain neuroscience education videos on YouTube, including to what extent the videos addressed (or negated) core target concepts of modern pain neuroscience education.

## MATERIALS & METHODS

### Search strategy

On 8th January 2018 we conducted three searches of YouTube.com using the following search terms: "pain education", "what is pain", and "pain brain" (see Fig. 1). Searches were conducted separately (rather than as a combined search with terms separated by 'OR') on one individual university-owned computer in Palo Alto (CA, USA). The three search terms were chosen based on discussion by the research team, which included clinicians and scientists working in the areas of chronic pain and pain neuroscience education, as well as young adult research interns who frequently used the YouTube site. We aimed to replicate a simple, naturalistic search strategy that could be conducted by lay consumers, including teenagers and older adults. Thus, we worked directly from the search list as well as observing and confirming that the first pages of 'recommended' videos from each search were included in the final results of our combined searches. Also, to replicate a natural search strategy, we did not restrict the search using either basic or advanced filters (for example "videos classified as 'educational'"), but instead allowed YouTube to sort video results by relevance according to the proprietary ranking algorithm in place on the search day. We did not clear the history of the computer prior to running the search or run searches in an incognito browser. At the time we conducted our search, YouTube made the first 590 videos in each search available to viewers by default, all of which (total = 1,770 video links) were added to a spreadsheet and then submitted to screening for duplicates

**Original Search**

Key words in the three searches were as follows: "pain education", "what is pain", and "pain brain"

**1770 videos (590 from each search) submitted to screening**

**Duplicates identified**

44 duplicates identified. 35 videos no longer available.

**1691 videos remaining**

**Inclusion criteria applied**

**Inclusion**

1. < 10 minutes in length
2. In English
3. Provides information on the neuroscience of pain

**Exclusion**

1. Video contained explicit content
2. Video *only* contained information on coping skills
3. Video *only* contained information on treatment

**106 videos remaining**

**Videos included in analysis**

1 video removed by YouTube in between analysis phases.

**106 videos** (full coding and analyses)
**105 videos** (coded for source and style)

**Figure 1** Flowchart of video search and screening.

and inclusion and exclusion criteria by the research team. The video URLs were used to identify videos for subsequent screening and coding.

We conducted an additional search for duplicate videos on 3rd October 2018. Specifically, we aimed to ensure that we had not missed any highly-viewed duplicates of particularly relevant videos to provide the most accurate summary of video characteristics (e.g., number of views). We decided to focus this step on those videos that had >10,000 views, and those that addressed at least one target concept of pain neuroscience education (see content coding below). These videos were accessed via their URLs, and the lists of 'suggested videos' from YouTube for each URL were searched. A new search was also conducted with the title of each video, and the first 50 recommended videos were briefly screened to ensure that none were duplicates. Where highly viewed duplicates were identified, we aggregated descriptive characteristics (i.e., view counts, likes, and dislikes) across all versions. If the videos had been split into multiple parts across different videos, we retained all separate videos rather than averaging descriptives.

Institutional Review Board (IRB) (ethics committee) approval was deemed unnecessary due to the nature of the study (see also *Fat et al., 2012*). The PRISMA checklist is included as Fig. S1.

## Inclusion and exclusion criteria

Inclusion criteria were as follows: (1) video length <10 min, (2) video was in English, (3) video included information regarding the neuroscience of pain. Exclusion criteria were as follows: (1) video contained explicit content, (2) video *only* contained content regarding how to cope with or manage pain (e.g., imagery or relaxation, coping skills), (3) video *only* contained information on treatment approaches to pain (e.g., physiotherapy exercises, psychological treatments, pharmacological treatments, or advertisements for individual clinics), (4) video made no reference to the role of the brain or central nervous system in the experience of pain.

Videos were initially screened for inclusion/exclusion by five members of the research team, with each video screened by one individual. Researchers who screened videos had the following qualifications: a faculty-level clinician-scientist specializing in chronic pain research and treatment, an early career experimental scientist specializing in chronic pain research, a masters-level research coordinator, and two undergraduate research interns. In an initial training phase, researchers who screened videos completed 10 h of training in which inclusion and exclusion criteria were discussed, and 10 practice videos were reviewed to ensure agreement in the screening approach. Screening was completed over a period of two weeks. Researchers were able to freely communicate throughout the screening process.

## Video characteristics

We aimed to describe the characteristics of all videos that met inclusion criteria. The majority of the video characteristics could be summarized using data provided by the YouTube website. To collate these data, we added all video links to three 'playlists' using the YouTube platform and used the scraper software package 'pafy' (https://github.com/mps-youtube/pafy), downloaded from GitHub and run in Python 3.6
on an iMac computer, to generate spreadsheets with the metadata for each video. For any videos for which the scraper did not return the metadata, video links were searched and data were extracted manually. We aimed to report: (1) number of views, (2) length in minutes, (3) duration on YouTube, (4) number of likes, (5) number of dislikes, and (6) YouTube category.

In addition to the video characteristics provided by YouTube, we also aimed to report the source and style of the videos. Two individuals (one study author (HL) and one collaborator) independently assessed all videos to create appropriate categories and to classify each video. The lead author (LCH) arbitrated all disagreements until final classifications were reached. Video styles were classified as follows: live action (talk to camera; role play; lecture; interview), animation (whiteboard; 2D; 3D), still images, screencast, and combination (e.g., live action + still images). Video source was determined by examining the person/organization who posted the video as well as any information given in the video itself about its creator. In some instances, the video had been clearly produced by a different source than the account it was posted by (e.g., an individual person sharing a TED Talk), in which case we classified the video according to the source from which it was produced. Video source categories were as follows: academic or medical research institution, healthcare company, educational organization, professional society, individual healthcare professional or academic, individual student, news broadcaster, or animation company. If the source could not be classified by viewing the video, by examining the profile of the individual/organization who posted the video, or by doing a brief search on Google.com of the individual/organization's name, the video source was classified as 'unclear.'

## Content analysis

We aimed to assess to what extent the videos that met inclusion criteria addressed core target concepts of pain neuroscience education according to the current state of the science and conceptual understanding in this field. We coded the videos for 7 Target Concepts of Pain Education that were previously generated at a summit held at Mount Lofty in Adelaide, Australia, in March 2018 (referred to hereafter as the Lofty Summit; Leake et al., in preparation). The seven Target Concepts of Pain Education were: (1) There are many potential contributors to anyone's pain; (2) We are all bioplastic; (3) Pain is not an accurate marker of tissue state; (4) Pain education is treatment; (5) Pain is a brain output; (6) Pain is a protector; (7) Pain can become overprotective/sensitized. The possible coding options for whether or not the videos met each target concept were as follows: *Yes, very well*; *Yes, but not very well*; *No, absent*; *No, contradicts*. The 'No, contradicts' code was used when the video provided information that directly contradicted a target concept (e.g., 'Pain is a measure of tissue damage' would directly contradict Target Concept 3). Individuals who attended the Lofty Summit, and thus who had been involved in detailed discussion of the target concepts, performed content coding for this study. Every video was coded by two separate individuals, using a coding form that was created in a spreadsheet and trialed by members of the research team before being distributed for data collection. Individuals who had been involved in the making of any of the videos (i.e., GLM) were

not allowed to act as content coders or to perform any analyses of video assessments. Our final content coding team comprised six individuals including one faculty-level clinician-scientist (clinical psychology) specializing in pediatric chronic pain research and treatment, one faculty-level scientist specializing in cognitive neuroscience and pain, one pediatric anesthetist specializing in women and children's health, two clinical physiotherapists and current PhD students researching pain education, and one divisional honors student researching chronic pain physical treatments. The lead author (LCH) oversaw all content coding activities and reviewed all videos. Content coding was completed at the individual institutions represented by each coder (five institutions across Australia and USA).

### Level of engagement

Individuals who performed content coding were also asked to indicate how engaging they thought the videos were (options: *Extremely engaging; Somewhat Engaging, Somewhat Boring; Extremely Boring)*. These additional data were considered highly subjective and were not included in our primary analyses. However, we considered that this additional information may be of use for clinicians who wish to select engaging videos to use in their clinical practice, and thus we incorporate this information into our description and discussion of the content coding results.

### Analytic approach

There were two content coders per video, thus a possible maximum of 742 matched codes across all seven target concepts. Altogether, there were 456 matched codes, i.e., where coders agreed on the degree to which the target concept had been addressed for each video. This reflects an absolute agreement level between coders of 62%. When arranging the options as an ordinal scale from '*Yes, very well*' to '*No, contradicts*', we found that the majority (90%) of the disagreements differed by only 1 point on the scale (e.g., one coder said '*Yes, very well*' and the other said '*Yes, but not very well*'). Given the relatively low level of absolute agreement, we analyzed and present the content coding data below in two ways: (1) a descriptive summary of all content codes to provide an overview of how the codes were represented across the full dataset; (2) a meta-critic approach in which content codes were assigned ordinal ratings (0 = '*No, contradicts*' or '*No, absent*'; 1 = '*Yes, but not very well*'; 2 = '*Yes, very well*') and summed to create total scores for each video, calculated individually across each target concept as well as across all target concepts.

## RESULTS

All raw data (content codes and characteristics), including the URLs to each video, are presented as (see Dataset S1).

### Initial search and screening for inclusion

The three search terms yielded 1,770 video links (590 video links per search; see Fig. 1). Many of the videos appeared in more than one search, and a number of videos were removed from the YouTube website shortly after the search. Altogether, there were 1,691 unique videos. Most of these videos ($N = 1,585$) did not meet inclusion criteria. The most

common reasons for exclusion were that the video contained no information about the neuroscience of pain (e.g., videos describing physiotherapy exercises or coping techniques such as distraction), the video was an advertisement for a pharmacological treatment or rehabilitation clinic, or the video included pain only as a metaphor for grief and anguish (e.g., there were a number of music videos about painful emotions). One additional video had been removed from YouTube between the time of the initial search and the compiling of videos for coding. Thus, altogether we found 106 unique videos that met the inclusion criteria.

## Video characteristics

Video characteristics are summarized in Tables 1 and 2 and provided in full in the (Dataset S1). Of note, one additional video was removed by YouTube before its source and style could be classified.

The most recently posted video was posted two days prior to the search, and the oldest video was posted over 11 years prior to the search ($Mdn$ = 2 years, 11 months). The majority of videos (74%) were posted within the last five years; 17% were posted in the last year.

Videos ranged in length from 41 s to 9 min and 54 s. The median video length was 3 min and 49 s. Most videos (77%) were five minutes or shorter.

Most videos were listed by YouTube under the Education category (44%), followed by People & Blogs (25%), and then Science & Technology (15%). When classifying the video sources (i.e., who made and/or posted the video), around one third of the videos (30%) were posted by an education company. Twenty videos (19%) were posted by an individual healthcare professional or academic, and 20 videos (19%) were posted by a healthcare company. Ten videos (10%) were posted by an academic or medical research institution. For seven videos (7%), the source was unclear.

About half of the videos (53%) were classified as Live Action, with the majority of those being an individual talking to the camera. About a quarter of the videos (26%) were animated, either using a whiteboard animation style or a 2D or 3D animation style. The rest of the videos with either still images, a combination of live action and still images, or screencasts.

Videos ranged from having four views to over five million views, with a median of 1,163 views. Just over half of the videos (52%) had been viewed over 1000 times, and 11 videos (10%) had been viewed over 100,000 times. The three most highly viewed videos accounted for 75% of total views. The most highly viewed video was 'The Science of Heartbreak', which was posted by the YouTube channel AsapSCIENCE in 2013 and had been viewed 5,307,048 times. The next two most highly viewed videos were both posted by TED-Ed ('How does your brain respond to pain? - Karen D. Davis', followed by 'How Do Pain Relievers Work? - George Zaidan'). These were also the most highly viewed videos when calculating views per day (i.e., number of views divided by duration on YouTube). All but one of the top 10 most highly viewed videos were listed under the Education or Science & Technology categories (the 10th most viewed video was listed under the People & Blogs

**Table 1  Characteristics of pain neuroscience education videos (from metadata; $n = 106$).** Percentages are rounded to the nearest whole number and thus totals may not add to exactly 100%.

| Characteristic | Values | Frequency | % |
|---|---|---|---|
| Duration on YouTube | 0–1 month | 1 | 1 |
| | 1 month–1 year | 17 | 16 |
| | 1–5 years | 60 | 57 |
| | 5–10 years | 26 | 25 |
| | 10 years+ | 2 | 2 |
| Video Length (mins) | 0–1 | 2 | 2 |
| | 2–5 | 80 | 75 |
| | 6–10 | 24 | 23 |
| YouTube Category | Education | 47 | 44 |
| | People & Blogs | 26 | 25 |
| | Science & Technology | 16 | 15 |
| | Nonprofits & Activism | 3 | 3 |
| | News & Politics | 3 | 3 |
| | Film & Animation | 3 | 3 |
| | Comedy | 3 | 3 |
| | How-To & Style | 2 | 2 |
| | Entertainment | 2 | 2 |
| | Sports | 1 | 1 |
| Number of Views | 0–100 | 16 | 15 |
| | 101–500 | 24 | 23 |
| | 501–1,000 | 10 | 9 |
| | 1,000–10,000 | 29 | 27 |
| | 10,000–100,000 | 16 | 15 |
| | 100,000+ | 11 | 10 |
| Number of "Likes" | 0 | 18 | 17 |
| | 1 | 8 | 8 |
| | 2–5 | 27 | 25 |
| | 6–50 | 29 | 27 |
| | 51–1,000 | 17 | 26 |
| | 1,001–5,000 | 4 | 4 |
| | 5,001+ | 3 | 3 |
| Number of "Dislikes" | 0 | 69 | 65 |
| | 1 | 7 | 7 |
| | 2–5 | 11 | 10 |
| | 6–50 | 12 | 11 |
| | 51–1,000 | 6 | 6 |
| | 1,001–5,000 | 1 | 1 |

**Table 2  Source and style of pain neuroscience education videos ($n = 105$).** We provide descriptives for the full category (in black) as well as individual classifications within a category (in grey). Percentages are rounded to the nearest whole number and thus totals may not add to exactly 100%.

| Characteristic | Categories | Frequency | % |
|---|---|---|---|
| Video Source | Education Company | 31 | 30 |
| ($n = 105$) | Individual Healthcare Professional or Academic | 20 | 19 |
| | Healthcare Company | 20 | 19 |
| | Academic or Medical Research Institution | 10 | 10 |
| | Unclear | 7 | 7 |
| | News Broadcaster | 5 | 5 |
| | Professional Society | 5 | 5 |
| | Animation/Media Company | 4 | 4 |
| | Individual Student | 3 | 3 |
| | | | |
| Video Style | Live Action | 56 | 53 |
| ($n = 105$) |    Individual talking to camera | 30 | 29 |
| |    Interview | 13 | 12 |
| |    Role play | 9 | 9 |
| |    Lecture | 4 | 4 |
| | Animation | 27 | 26 |
| |    Other | 15 | 14 |
| |    Whiteboard | 12 | 11 |
| | Still Images | 11 | 10 |
| | Combined (Live Action + Images) | 9 | 9 |
| | Screencast | 2 | 2 |

category). Animated videos had a far greater average number of views ($Mdn = 10,109$ views) than non-animated videos ($Mdn = 881$ views).

Almost all of the videos (83%) received at least one "like" rating by viewers ($Mdn = 6$; $Range = 0–68,175$). Almost two thirds of the videos (60%) received over five "like" ratings, and seven videos (7%) received over 1000 "likes". Most videos (65%) did not receive a "dislike" rating, although 19 videos (18%) received over five "dislikes", and one video received over 1000 "dislikes".

## Content analysis

Content codes are summarized in Fig. 2 and are provided in full in the (Dataset S1).

The compilation of all codes is presented in Fig. 2A. '*No, absent*' was the most commonly used code, indicating that most videos did not address the target concepts of pain neuroscience education. However, the use of the 'Yes' codes indicated that all seven target concepts were addressed to some extent across the 106 videos. There code '*No, contradicts*' was used very infrequently, and this code was not used at all for two of the target concepts, indicating that there were relatively few instances of misinformation across the videos. The target concept 'Pain is a brain output' received the most '*Yes, very well*' codes but also the most '*No, contradicts*' codes, indicating that there was variation in the extent to which different videos addressed this target concept. 'We are all bioplastic'

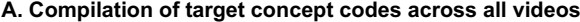

**A. Compilation of target concept codes across all videos**

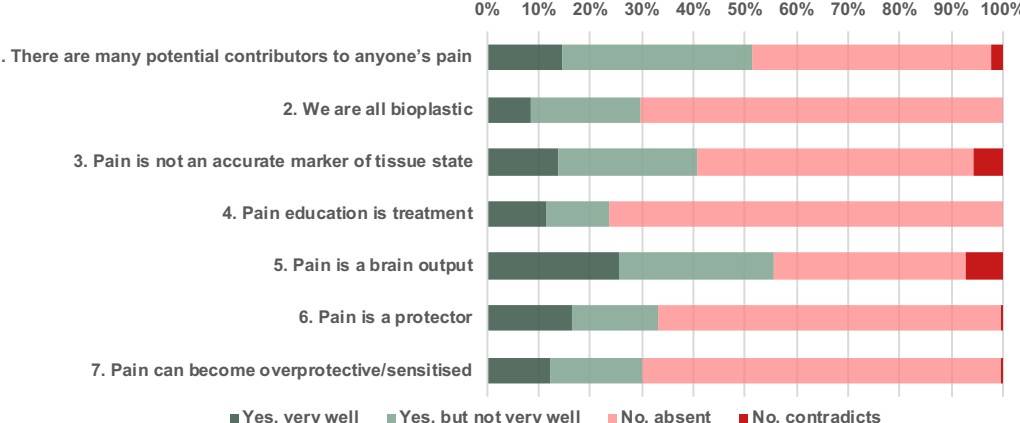

**B. Number of videos that addressed each target concept 'very well'**

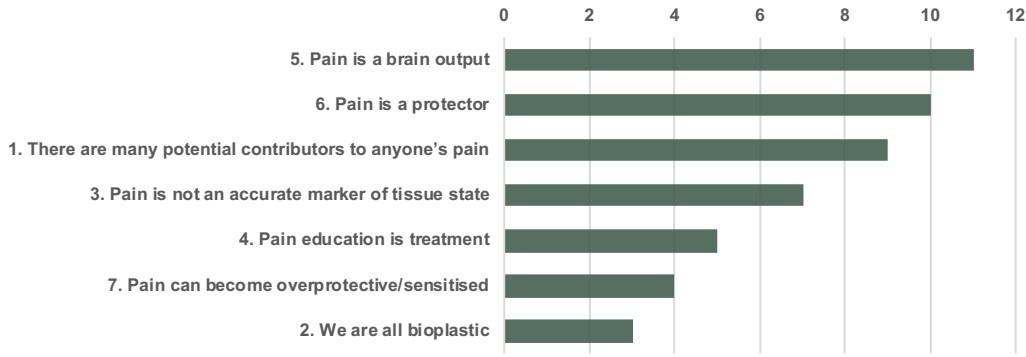

**C. Number of target concepts that were addressed 'very well' across all videos**

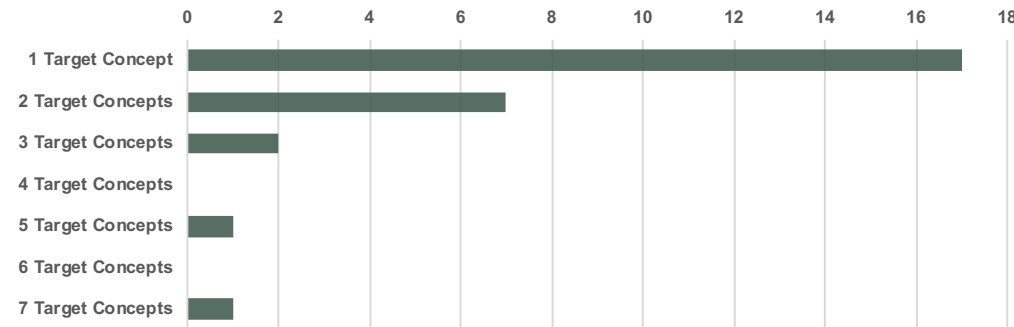

**Figure 2** **Content coding results for the seven target concepts of pain neuroscience education.**

received the least '*Yes, very well*' codes. The target concept 'Pain education is treatment' received the most '*No, absent*' codes.

Next, Figs. 2B and 2C illustrate results of the meta-critic approach. Figure 2B displays the number of videos that achieved the highest possible score across each target concept, representing that both coders agreed that the target concept was addressed very well. The target concept 'Pain is a brain output' was considered to be well addressed by the most videos

($N = 11$ videos), followed by 'Pain is a protector' ($N = 10$ videos). The target concepts 'We are all bioplastic' ($N = 3$ videos) and 'Pain can become overprotective/sensitized' ($N = 4$ videos) were well addressed by the least videos. As can be seen in Fig. 2C, there were 17 videos that addressed at least one target concept very well. Relatively few videos addressed 2-6 of the target concepts very well. Only one video addressed all seven target concepts very well ('Tame the Beast: It's Time to Rethink Persistent Pain').

When considering to what extent content coders found the videos engaging, we found nine videos that were considered to have addressed at least one target concept very well (i.e., a perfect content score summed across both coders) and were considered to be engaging (i.e., an engagement score of at least five out of six summed across both coders). These videos ranged from two and a half minutes to just under eight and a half minutes. These nine videos, including their URLs, are highlighted in the (Dataset S1).

## DISCUSSION

Our objective was to examine the availability, characteristics, and content of YouTube videos addressing the neuroscience of pain. Most of the videos that resulted from our search did not meet inclusion criteria. A large portion were advertisements for pharmacological treatments, rehabilitation clinics, or physiotherapy exercises that did not contain information about the neuroscience of pain. Nonetheless, we found 106 unique videos that met inclusion criteria. Almost three times as many videos that met inclusion criteria were posted within five years prior to the search compared to those posted 5–10 years prior to the search, highlighting the increasing use of YouTube as a tool for disseminating information about pain. The most recent video was posted just two days prior to the search. However, the majority of the videos that we identified as addressing at least one target concept of pain neuroscience education very well were posted over three years prior to the search, indicating a relative paucity of newer videos that may be particularly useful for delivering pain neuroscience education.

The videos that met inclusion criteria varied greatly in their source and style. The majority of videos were appropriately categorized by YouTube as 'Education' or 'Science & Technology', although a number of the videos that we identified as addressing target concepts of pain neuroscience education were categorized by YouTube under additional categories (e.g., the animated series of videos posted by the illustrator 'Brainman' were categorized as 'Nonprofits & Activism'). This indicates that restricting a YouTube search to just the 'Education' category may miss some useful videos. A small number of videos were clearly marked as originating from a reputable source such as an academic or medical research institution (10%) or a professional academic medical society (5%). Almost 20% of the videos were posted by an individual healthcare professional or academic. Videos were also posted by educational and media organizations, and in some cases these organizations directly reported working with academic or healthcare professionals in the development of the video in its description. However, for other videos, it was unclear to what extent these independent organizations had collaborated with experts to develop the video content. On the whole, there was very little information posted on the YouTube website about
how or with whom the video content was developed, making it difficult for (especially non-academic) viewers to assess the reputation of the video sources. In a small number of cases (7%), despite conducting an additional search on Google.com of the name of the individual or organization who posted the video, the source itself was unclear and could not be categorized.

The videos also varied greatly in their level of engagement by users. Some videos were viewed less than 20 times, while others were viewed over one million times. Two of the most highly viewed videos were posted by the educational organization TED-Ed. Given that academic and healthcare professionals can work with TED-Ed and its animators to create highly engaging video content, this platform is particularly promising for delivering future information on the neuroscience of pain. Indeed, animated videos had a far greater average number of views than non-animated videos, indicating that this style of video may be particularly useful to engage viewers.

The majority of videos did not address any of the seven target concepts of pain neuroscience education. However, based on absolute agreement between coders, we found 17 videos which coders agreed addressed at least one target concept very well. Only one video addressed all seven target concepts very well ('Tame the Beast: It's Time to Rethink Persistent Pain'). As can be seen in Fig. 2, the target concept 'Pain is a brain output' was considered to be well addressed by the most videos. The target concepts 'We are all bioplastic' and 'Pain can become overprotective/sensitized' were well addressed by the least videos, indicating the need for more video resources specifically addressing these concepts. Interestingly, there were very few instances in which coders identified information that directly contradicted the target concepts. This appears to be somewhat in contrast to previous studies that have identified a worrying amount of misinformation in health-related YouTube videos (Madathil et al., 2015; Loeb et al., in press). However, it is likely that our findings are at least in part due to the fact that we excluded videos which did not contain any information on the role of the brain and/or central nervous system in pain (see Methods: Inclusion and Exclusion Criteria). This criterion by default would likely have excluded videos promoting a peripheral-centric view of the science of pain, for example arguing that pain is a direct consequence of tissue damage (which would have contradicted the third target concept, 'Pain is not an accurate marker of tissue state'). It remains unclear to what extent the excluded videos, which users may still access and view for pain-related information, promoted misinformation about pain.

One potential outcome of our review is identifying videos that could be used for delivering pain neuroscience education as part of interdisciplinary clinical practice. With this in mind, it is important to consider how one should judge whether or not a video could indeed be considered useful in this regard. While we found only one video that coders agreed had addressed all seven target concepts very well, videos that addressed only one or two target concepts may be equally or more useful, especially if they address a gap in the individual patients' conceptualization of pain, or a concept that the clinician believes is most central to his/her treatment approach. We also focused on seven target concepts that emerged from a recent summit on this topic (see Methods: Content Analysis), but acknowledge that there are other target concepts that we did not code for but could also

be relevant. Moreover, there is currently little understanding of which, if any concepts are most important, nor of the necessary dose–response relationship of delivering such target concepts. Another important consideration is which videos will be most engaging for patients. One potentially relevant metric for this is the number of views on YouTube. To provide an exhaustive review of available videos, we decided to include videos that contained any information related to the neuroscience of pain, even if this was not the exclusive focus of the video. In that regard, the most highly viewed video ('The Science of Heartbreak'—posted by AsapSCIENCE) focused on pain as a metaphor for grief and anguish, and thus would likely not be a candidate for use in clinical practice for individuals with persistent pain. The second and third most viewed videos ('How does your brain respond to pain?—Karen D. Davis' and 'How Do Pain Relievers Work?—George Zaidan'), however, would likely be more appropriate, although they may be most useful for certain populations (e.g., those considering taking pain medications). We also asked our team of reviewers who conducted content coding to rate how engaging they found the videos. Those data reflect only the opinions of the small reviewer team, all of whom are actively working in consumer education, pain research, or clinical practice, and we provide this information only as a guide for videos that may be particularly useful for clinical practice. However, an important next step will be to collect engagement data from patients themselves, for example through focus groups or user surveys. It will be particularly important to recruit different age groups, including children, adolescents, and younger and older adults.

Our study is one of a growing number of attempts to systematically review health-related video resources on YouTube. Previous studies have reviewed videos with information about myocardial infarction (*Pant et al., 2012*), dental education (*Knosel, 2011*), and smoking cessation (*Backinger et al., 2011*). There are also a small number of published studies specifically reviewing pain-related information on YouTube, including a review of videos on pain management practices during infant immunization (*Harrison et al., 2014*), newborn blood tests (*Harrison et al., 2018*), and caregiver cancer pain management (*Wittenberg-Lyles et al., 2014*), as well as videos providing broader information about painful disorders such as arthritis (*Singh, Singh & Singh, 2012*). There is also a larger literature reviewing Internet-based resources, including websites and apps (*Corcoran et al., 2009*; *Rosser & Eccleston, 2011*; *Bailey et al., 2013*; *De La Vega & Miró, 2014*; *Lalloo et al., 2015*; *Smith et al., 2015*), for their usefulness and appropriateness for people with pain. Taken together, this literature indicates that people living with persistent pain are increasingly using the web to find information (*De Boer, Versteegen & Van Wijhe, 2007*; *Ziebland, Lavie-Ajayi & Lucius-Hoene, 2014*) and that there are a growing number of online resources available to patients seeking information about pain, including websites providing written information about pain disorders, and tools (e.g., apps) that allow more active tracking of pain and that teach coping skills. Our study shows that there are also a small number of potentially useful videos that deliver pain neuroscience education, that can be freely accessed on YouTube.com. Our study also highlights the challenge of conducting a systematic review on the YouTube site. Unlike traditional systematic reviews, online information can be uploaded or removed at any time, and even simple searches can yield different results depending on the user's search history, their geographic location,

and fluctuating popularity of the video content (*Sampson et al., 2013*). Indeed, a number of videos were removed from the YouTube site in between each stage of our search and coding the videos, highlighting the dynamic nature of the online platform and the necessity for flexible approaches to designing and conducting systematic reviews in this area.

This study has a number of limitations, providing directions for future research. First, we only included videos that were under 10 min long and were in English. This inclusion criterion was based primarily on the assumption that shorter videos would be more useful for clinical practice, and more likely to be watched in full by patients seeking information on YouTube. Whilst this is in line with previous systematic reviews of YouTube (e.g., *Steinberg et al., 2010*), we likely missed longer videos that might have offered excellent content (e.g., *Stanford Children's Health | Lucile Packard Children's Hospital Stanford, 2017*) or could be used in non-English speaking countries (e.g., *Deutsches Kinderschmerzzentrum, 2014*). Second, unlike in traditional systematic reviews using bibliometric services such as PubMed or Ovid MEDLINE, we were not able to include MESH terms in our searches and therefore likely missed useful videos that used disorder-specific terminology instead of the word 'pain' (e.g., *Deutsches Kinderschmerzzentrum, 2016*). This is especially relevant given that individuals are likely to search for information about their specific diagnoses. Future reviews of disorder-specific terms would be particularly useful. Third, we limited our search to the YouTube website. Future studies could expand their search to additional video-hosting platforms such as Vimeo and Yahoo Video, given the growing popularity of these additional sites. This approach could help to identify additional videos as well as providing a more accurate assessment of user engagement by combining the number of views for the same videos across platforms. In addition, videos could have also been hosted and viewed on independent websites. For example, two of the videos we included in analyses were also hosted on TED-Ed.com, where they received a larger number of views than on the YouTube site. We thus likely underestimated user engagement with these videos (although they were still identified as two of the most highly viewed videos from our search). Relatedly, we also only summarized views on the YouTube site itself, with no mechanism to count how many times videos were downloaded and shared on private or closed-loop networks, or offline. Fourth, absolute agreement between individuals who performed content coding was relatively low. This was somewhat surprising given that we had purposefully recruited individuals who attended the Lofty Summit to perform content coding, and therefore were considered to have shared expertise in their understanding of the seven target concepts. Of note, this approach resulted in an inherent bias in that the seven target concepts coded in this review were developed by some of the study authors, and other study teams may have selected to code for different concepts. Nonetheless, rather than attempt to reach a shared consensus, we chose two analytic approaches that we considered best reflected the range of the data (i.e., describing all codes) and conservatively summarized the data on which coders were in absolute (positive) agreement (i.e., where both coders agreed that the target concepts had been addressed very well). Note that all data can be inspected in the (Dataset S1). Future studies employing a similar approach would benefit from conducting a more extensive training period for all coders in order to reach higher levels of agreement, as well as developing a more detailed coding scheme.

Fifth, we did not use a validated tool to assess the quality of information in the YouTube videos. There are a number of tools that have been developed for evaluating the quality of written information online (e.g., the Quality Website Index (QWI), Health on the Net Foundation Code of Conduct (HONcode), and the DISCERN instrument). None of these tools were deemed appropriate for this review, primarily because a number of criteria to indicate quality either could not be assessed (e.g., quality of language and written material) or were not appropriate for video platforms (e.g., certification of website standards). Nonetheless, adapted versions of these tools may be useful for future studies. Such tools may include assessment of sound and picture quality as well as content. The systematic review of non-traditional scientific content (e.g., videos instead of peer-reviewed papers) is an emerging methodology and, in its early stages, will be subject to these significant limitations until the field grows and appropriate tools are developed and validated

## CONCLUSIONS

In conclusion, YouTube is an increasingly popular source for accessing consumer health information, including information about pain. Pain neuroscience education is now recommended in guidelines as part of an interdisciplinary approach to treating persistent pain, and videos are one method for delivering educational information in an engaging way that can be easily distributed. We found that a large variety of videos with information about pain are available on YouTube, and some of these videos had been viewed over a million times, indicating that users are engaging with pain-related content online. Animated videos were typically more highly viewed than non-animated videos, and thus animation may be a particularly useful medium for delivering pain neuroscience education. However, only a small number of videos were deemed to address target concepts of pain neuroscience education and were considered to be engaging. Additional research is needed to assess to what extent patients find the videos engaging and whether they can be effectively used to promote both effective learning and behavior change.

## ACKNOWLEDGEMENTS

We thank Nele Loecher for her assistance in classifying video characteristics.

### Funding
The authors received no funding for this work.

### Competing Interests
Author GLM was involved in the development of a number of videos that were included in this review. GLM has received support from Pfizer, Kaiser Permanente, workers' compensation boards in Australia, Europe & North America, the Port Adelaide Football Club, Arsenal Football Club. He receives royalties for books on pain and speaker fees for lectures on pain.

## Author Contributions

- Lauren C. Heathcote conceived and designed the experiments, performed the experiments, analyzed the data, prepared figures and/or tables, authored or reviewed drafts of the paper, approved the final draft, contributed to analytic approach.
- Joshua W. Pate analyzed the data, prepared figures and/or tables, authored or reviewed drafts of the paper, approved the final draft, contributed to analytic approach.
- Anna L. Park conceived and designed the experiments, performed the experiments, approved the final draft.
- Hayley B. Leake analyzed the data, prepared figures and/or tables, authored or reviewed drafts of the paper, approved the final draft, contributed to analytic approach.
- G Lorimer Moseley authored or reviewed drafts of the paper, approved the final draft, contributed to analytic approach.
- Corey A. Kronman performed the experiments, approved the final draft.
- Molly Fischer conceived and designed the experiments, performed the experiments, approved the final draft.
- Inge Timmers performed the experiments, analyzed the data, prepared figures and/or tables, authored or reviewed drafts of the paper, approved the final draft, contributed to analytic approach.
- Laura E. Simons conceived and designed the experiments, performed the experiments, authored or reviewed drafts of the paper, approved the final draft, contributed to analytic approach.

## Data Availability

Pain neuroscience education on YouTube: a systematic review

Lauren C. Heathcote, Joshua W. Pate, Anna L. Park, Hayley B. Leake, G Lorimer Moseley, Corey A. Kronman, Molly Fischer, Inge Timmers, Laura E. Simons

bioRxiv 492967; doi: https://doi.org/10.1101/492967.

## Supplemental Information

Supplemental information for this article can be found online at http://dx.doi.org/10.7717/peerj.6603#supplemental-information.

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
