# Peer review of "Pain neuroscience education on YouTube"

_PeerJ, doi:10.7717/peerj.6603_

## Round 0.1 · original submission · Minor Revisions

Dear Authors,

Please read the comments of the peer reviewer comments and do the necessary corrections and improvements, in particular those of Reviewer 1

·

Basic reporting

Thank you for the opportunity to review the article. This article is a review of YouTube on the use of pain neuroscience education videos posted on YouTube. Raw data is supplied and structure is appropriate. The intro and background support the need and build to the purpose of the study. The writing is of good quality.

Experimental design

This is an interesting topic that has not been previously looked at. The methodology to review YouTube is well explained and sound. I do have some concerns with the experimental design and the selection of the quality measurement tool. The authors do state in the limitations of the study that the tool has not been validated and that the absolute agreement between individuals was relatively low. These two limitations are very significant flaws within the design. A non-validated tool to assess quality and poor reliability with the use of the tool lead to significant concerns with any of the findings derived from the tool regarding quality of the videos. Additional concern is with potential bias of the researchers around the tool and their involvement with the Lofty Summit and creation of the seven target concepts of PNE. I recognize there may not be a better tool out there to assess quality. Nevertheless, this tool may not be ready yet for such purposes even if it is the best that is currently available. Because of the non-validation and risk for high bias around the quality assessment tool, the results and reporting of the quality of the videos based on these criteria can be construed as bias as well. In addition, the author(s) involvement with some of the YouTube videos on the list and creation of the seven target concepts was not provided in the text, this should be listed as a limitation and potential conflict of interest within the text for the reader to have full transparency of the authors and their involvement.

Validity of the findings

Significant concerns around the validity of the findings in Figure 2 in relation to the seven target concepts and the use of them for quality assessment measurement as previously stated. The authors will need to provide more significant justification on the validity of the use of these target concepts along with their bias for them and why other tools that have been validated and used were not appropriate. There is no concern with the characteristics data provided in table 1 and 2.

Additional comments

The methodology of carrying out and searching for videos within YouTube is appropriate along with the reporting of the results around the characteristics reported. The main concern is around the use of the quality assessment tool and any findings associated with this tool as previously stated.

·

Basic reporting

This is a well-conducted review with appropriate context, good structure and well-written. In the absence of guidelines for this type of review, you have conducted a methodologically strong review.

Experimental design

No comment

Validity of the findings

This is a well conducted review. Only one error found in the discussion of results, where table 2b should be referred to as table 2 c and table 2 c should be referred to as table 2b.

Additional comments

Well done on a challenging area to review. This is a very novel review which I think will be very useful and interesting to both clinicians and patients.

Reviewer 3 ·

Basic reporting

-Line 72: has some harsh phrasing (most burdensome health condition facing humankind)

-On a linguistic manner: line 86: sentence is hard to follow

-The following sentences are somewhat unclear to the reader: ‘Across all videos and target concepts, the majority of the codes were ‘No, absent’, indicating that most videos did not address the target concepts of pain neuroscience education. However, there were a small number of ‘Yes, very well’ and a slightly larger number of ‘Yes, but not very well’ codes, indicating that all seven target concepts were addressed to some extent across the 106 videos.’

-You could consider to shorten the first part of the discussion between lines 363 and 464, as even though it is indeed relevant it is somewhat abundant described.

Experimental design

1. I found the search terms youtube to be slightly low in quantity? And could you explain how the search on one computer might have influenced the outcome?
2. Could you explain why you did not use snowballing as well? As that would be what persons/patients naturally would do
3. As our history influences our searches, could you explain why this was not cleared and furthermore, why you did not use an ignocnito browser?
4. Could you explain why you had one search moment?
5. One of the inclusion criterias is that the video is below 10 minutes, what is the reason for this?
6. Could you explain in more detail what ‘no contradicts’ means; does the video actively state that the specific concept of pain education is not true?
7. Did you address the quality of the videos in terms of sound/picture quality/clearness of voice/english/etc.?
8. One of the issues I have is with the many different reviewers you used in the content coding, how did you ensure you all had the same vision/quality?
9. Why the use of an absolute agreement level instead of Cohen’s Kappa for agreement?
10. Furthermore, I wonder what your Engagement criteria where?

Validity of the findings

Corresponds with 2.

Additional comments

First of all I would like to thank you for your contribution to the field of Pain Neuroscience Education. With this extensive research you have highlighted an important consequence of the popularity of PNE, namely the translation of institutes and (semi)professionals of PNE into videos.

Furthermore, you have written an academic article that is clear and readable for our fellow healthcare professionals working with patients with chronic pain.

However, prior to publishing it, I would like to offer you some suggestions for improvement as listed below.

---

## Round 0.2 · accepted · Accept

Dear Authors,We are happy to accept your revised manuscript.Thank you.

# ·

Basic reporting

No comment. Authors made necessary changes and addressed concerns from initial review.

Experimental design

No comment. Authors made necessary changes and addressed concerns from initial review.

Validity of the findings

No comment. Authors made necessary changes and addressed concerns from initial review by further addressing limitation of reliability of target concepts.

Additional comments

Authors made necessary changes and addressed concerns from initial review. Thank you for your contribution.